# Sperm rDNA Copy Number and Methylation Are Associated with Male-Factor Infertility

**DOI:** 10.3390/ijms262110657

**Published:** 2025-11-01

**Authors:** Alina Michler, Sarah Kießling, Jana Durackova, Thomas Hahn, Martin Schorsch, Thomas Haaf

**Affiliations:** 1Institute of Human Genetics, Julius Maximilians University, 97074 Würzburg, Germany; alina.michler@stud-mail.uni-wuerzburg.de (A.M.); sarah.kiessling@stud-mail.uni-wuerzburg.de (S.K.); jana.durackova@uni-wuerzburg.de (J.D.); 2Institute of Clinical Genetics and Genomic Medicine, University Hospital, 97080 Würzburg, Germany; 3Fertility Center, 65189 Wiesbaden, Germany

**Keywords:** absolute rDNA copy number, active rDNA copy number, ART outcome, idiopathic male infertility, semen parameters, sperm epigenome

## Abstract

Both absolute and presumably active rDNA (with a hypomethylated promoter region) copy number (CN) in the haploid human sperm genome are highly variable among individuals. Using a combination of droplet digital PCR and deep bisulfite sequencing, we have quantified absolute and presumably active rDNA CN in sperm samples (*N* = 190) with normal (NSPs) vs. abnormal semen parameters (ASPs), as well as in samples leading or not leading to a clinical pregnancy. ASP samples had a significantly lower presumably active CN (104 ± 31) than normozoospermic samples (115 ± 31). The loss of presumably active rDNA copies is explained by an increased promoter methylation (13.9% in ASP vs. 12.1% in NSP). When correcting for confounding factors, most importantly semen quality, samples not leading to a clinical pregnancy after IVF/ICSI displayed a significantly lower absolute (225 ± 51) and presumably active CN (103 ± 30) than samples with pregnancy (249 ± 62 and 115 ± 31, respectively). This between-group difference was most noticeable in normozoospermic males: absolute CN 220 ± 54; presumably active CN 107 ± 32 in samples without pregnancy and absolute CN 246 ± 63; presumably active CN 120 ± 28 in samples with pregnancy. We propose that absolute/active rDNA CN in sperm is a modulating factor contributing to idiopathic male infertility. In NSP samples, presumably active CN increases with absolute CN, which may have a positive impact on fertility and ART outcome. Our results suggest that approximately 60 active sperm rDNA copies are sufficient to establish a pregnancy.

## 1. Introduction

In Europe and other highly developed countries, the 12 month prevalence of infertility ranges from 3.5% to 16.7%, compared to 6.9% to 9.3% in less developed countries [1,2]. Approximately half of the cases involve male-factor infertility. Overall, 5–10% of men are estimated to be infertile. Male infertility may be due to abnormal sperm function and quality, hormonal and other disorders, and lifestyle factors including age, smoking, and obesity. Classical genetic causes for male infertility are chromosome aberrations, azoospermia factor deletions, and mutations in the *CFTR* gene. Although exome sequencing has identified a growing number of genes associated with male infertility [3], the majority of cases remains unexplained (idiopathic infertility) and may be multifactorial in origin.

In addition to genetic factors, the sperm epigenome, which is programmed in the male germline and affected by environmental factors, may have an impact on fertility and embryo development [4,5]. Aberrant methylation patterns in developmentally important genes, in particular, imprinted genes, and in repeats have been associated with idiopathic male infertility [6,7,8]. Age-related sperm methylation changes preferentially occur in genes and biological processes associated with development [9,10,11]. An age-related gain of ribosomal DNA (rDNA) methylation has been observed in human somatic tissues [12,13] as well as in germ cells [14,15,16].

In the diploid human genome, several hundred rDNA transcription units (TUs) are arranged in tandem in the nucleolus organizer regions of the short arms of acrocentric chromosomes. In our previous studies, the absolute number of rDNA TU ranged from 243 to 895 (469 ± 107) in the blood of adult and old (29–71 years) individuals [13] and from 98 to 404 (219 ± 47) in haploid sperm of donors from 29 to 72 years of age [16]. The rRNA transcribed from active rDNA copies is a prerequisite for ribosome biogenesis and protein production. Despite the importance of rDNA for basically all biological processes, the functional consequences of copy number (CN) variation among normal individuals are poorly understood. It has been suggested that rDNA CN acts as a modulator of multifactorial disease [17]. For example, absolute rDNA CN have been associated with schizophrenia [18], body mass index [19], blood cell composition, and renal function [20].

To compensate for the enormous rDNA CN variation, 40–70% of all rDNA TU (depending on absolute CN) are transcriptionally silenced by rDNA methylation [21,22]. The increasing rDNA methylation with age in both germline and somatic cells is due to the age-related loss of hypomethylated (≤10%) presumably active and a gain of methylated (>10%) presumably inactive rDNA copies. Because of declining sperm quality, the chances of achieving a pregnancy are reduced with advanced paternal age [23,24], and this is associated with increasing rDNA methylation [14,16].

The aim of this study was to find possible associations of rDNA CN and methylation with sperm quality and pregnancy success after in vitro fertilization (IVF)/intracytoplasmatic sperm injection (ICSI). To this end, we compared absolute and presumably active rDNA CN between sperm samples with normal (NSPs) and abnormal semen parameters (ASPs), as well as between samples with positive or negative IVF/ICSI outcomes. Our results enhance existing knowledge on the (epi)genetic factors contributing to male infertility and form a basis for future studies to establish rDNA CN as a prospective marker in assisted reproduction.

## 2. Results

### 2.1. Association of Presumably Active rDNA CN and Semen Parameters

In order to study the interrelation between rDNA CN and male infertility, we recruited 94 NSP (Table 1) and 96 ASP sperm samples (Table 2). Semen and clinical parameters of the analyzed samples can be found in Appendix A. All samples were from men attending a fertility center. Both groups were matched in age and body mass index (BMI). There was no between-group difference in absolute rDNA CN (236 ± 61, median 223 in the NSP and 240 ± 56, median 230 in the ASP group) (Figure 1A). However, there was a significantly (*p* = 0.001) lower promoter methylation in the NSP (12.1 ± 3.2%, median 11.7%), compared to the ASP (13.9 ± 3.6%, median 13.7%) group (Figure 1B). This is consistent with a significantly (*p* = 0.004) higher presumably active rDNA CN with a hypomethylated promoter in the NSP (115 ± 31, median 116) than in the ASP group (104 ± 31, median 101) (Figure 1C).

When combining both groups (*N* = 190), absolute CN was not significantly correlated with semen parameters (concentration: *ρ* = −0.04, *p* = 0.56; motility: *ρ* = −0.10, *p* = 0.18); morphology: *ρ* = −0.06, *p* = 0.45). In contrast, there was a significant correlation of active CN with sperm concentration (*ρ* = 0.14, *p* = 0.05) (Figure 2A) and morphology (*ρ* = 0.16, *p* = 0.03) (Figure 2B), but not with motility (*ρ* = 0.10, *p* = 0.15). As shown previously [16], active rDNA CN significantly decreased with donor age (Figure 3A), whereas the BMI had no detectable effect (Figure 3B).

To corroborate the association between promoter methylation/active CN and abnormal semen parameters, we have compared NSP samples with different ASP subgroups (Appendix A). Most ASP samples exhibited oligoasthenoteratozoospermia (OAT) (*N* = 37) or asthenozoospermia (reduced motility) (*N* = 49). There was a significant or trend difference in methylation (OAT: *p* = 0.023; asthenozoospermia: *p* = 0.002) and presumably active CN (*p* = 0.058 and *p* = 0.032, respectively), compared to the NSP group. Despite small sample sizes, significant or trend between-group differences were also observed for oligozoospermia (low sperm count), cryptozoospermia (more severe variant of oligozoospermia), and teratozoospermia (high percentage of abnormally shaped sperm). As expected, absolute CN did not differ between NSP and ASP subgroups.

### 2.2. rDNA CN and ART Outcome

Following IVF/ICSI, 100 sperm samples achieved a pregnancy and 89 did not. Absolute CN was significantly (*p* = 0.006) higher in samples leading to a pregnancy (249 ± 62; median 243) than in samples not leading to pregnancy (225 ± 51; median 220) (Figure 4A). Promoter methylation was comparable in both groups (12.7 ± 3.2%, median 12.0% in samples with and 13.3 ± 3.8%, median 12.5% in samples without pregnancy) (Figure 4B). Similarly to absolute CN, presumably active CN was significantly (*p* = 0.006) higher in samples with (115 ± 31; median 110) than in samples without pregnancy (103 ± 30; median 102) (Figure 4C).

It is not unexpected that semen parameters were significantly (concentration: *p* = 0.003; motility: *p* = 0.030; morphology: *p* < 0.001) better in the group with pregnancy (Table 3), compared to samples not leading to a pregnancy (Table 4). Therefore, we built a statistical regression model controlling for the effects of semen parameters, donor age, and BMI on pregnancy outcome. Even when considering these possible confounding factors, the between-group difference in absolute (*p* = 0.016) and presumably active CN (*p* = 0.015) remained significant.

The lowest numbers of hypomethylated, presumably active rDNA copies in all sperm samples were 43, 47, 50, 55, and 56. In sperm samples leading to a pregnancy, the lowest observed numbers were 61, 65, 67, and 69 (2×). This is a significant between-group difference (*p* = 0.009), suggesting that approximately 60 hypomethylated copies are sufficient to establish a pregnancy.

Moreover, we analyzed the effects of rDNA CN on pregnancy outcome in the NSP and the ASP groups separately. In males with normal semen parameters (Figure 5A), both absolute CN and presumably active CN were significantly (*p* = 0.027 and 0.036, respectively) higher in samples with pregnancy (absolute CN 246 ± 63, median 234; presumably active CN 120 ± 28; median 119) than in samples without pregnancy (absolute CN 220 ± 54, median 212; presumably active CN 107 ± 32; median 106). In males with abnormal spermiogram (Figure 5B), absolute and presumably active CN were also higher in samples leading to a pregnancy (absolute CN 254 ± 62, median 247; presumably active CN; 109 ± 33; median 102), compared to samples that did not (absolute CN 229 ± 50, median 220; presumably active CN 100 ± 28; median 100); however, this between-group difference was only trend-significant (for absolute CN) or not significant (for presumably active CN). Collectively, our data suggest that an increased absolute and presumably active CN in sperm is associated with higher chances to achieve an IFV/ICSI pregnancy, in particular, for men with normal semen parameters.

## 3. Discussion

### 3.1. Presumably Active rDNA CN as a Modulatory Factor in Male Infertility

Idiopathic male infertility is not caused by highly penetrant genetic or chromosome mutations but is rather a multifactorial disorder with numerous genetic, epigenetic, and environmental factors, all of small effect size, contributing to the phenotype. Fertility problems may occur when the number of adverse factors exceeds a critical threshold. Sperm samples with abnormal semen parameters or samples not leading to a pregnancy after IVF/ICSI are endowed with a significantly lower active rDNA CN than samples with normal semen parameters or clinical pregnancy. We estimate that in humans, 60 active rDNA copies are sufficient for normal sperm function. Classical studies in *Drosophila* [25,26] have shown reduced fertility of flies with a 50% reduced rDNA CN. Chicken embryos endowed with less than 50% of the normal rDNA CN were arrested in development [27], whereas chicken lines with increased rDNA CN grew faster than controls [28].

The reduced number of presumably active copies with a hypomethylated promoter in samples with abnormal spermiogram is mainly caused by increased promoter methylation. Absolute CN are comparable in the NSP and the ASP group. Previously, we have shown that sperm ALU repeats are hypermethylated in men with abnormal semen parameters, compared to normozoospermic males [6]. This possibly indicates a broader context between aberrant repeat methylation in the male germline and poor semen quality.

Samples without pregnancy exhibited a reduced absolute and presumably active CN, whereas the promoter methylation level was comparable with samples leading to a pregnancy. This is consistent with our earlier study [16] showing a strong positive correlation between presumably active (hypomethylated) CN and absolute CN in human sperm. In men with normal semen parameters, an increased absolute and, consequently, presumably active CN may have a positive impact on ART outcome. Taken together, our study promotes the idea that low absolute and presumably active CN are involved in the etiopathogenesis of male infertility. However, the mean between-group difference (with or without pregnancy) is only 10–20 copies and still within the normal range of methylation variation.

### 3.2. Possible Role for Presumably Active Sperm rDNA Copies in Early Development

The sperm epigenome is the result of a genome-wide demethylation and remethylation cycle in the male germline. The sperm methylation patterns are completed at the latest in pachytene spermatocytes [29,30]. Accumulating evidence suggests that sperm genes with an unmethylated promoter region are prone to transcriptional activation in early embryo development [5,31,32,33]. In this light, the number of active sperm rDNA copies may affect embryonic genome activation (EGA), which is essential for the establishment of totipotency [34]. In human embryos, the rDNA transcription starts after the four-cell stage [35] at the beginning of EGA, which is initiated at least two days after fertilization at the 4–8 cell stage [36]. Hypomethylation of sperm rDNA genes, which are transmitted through fertilization into the zygote, may be a prerequisite for their transcription during preimplantation development. In the mouse model, experimental inhibition of rDNA transcription interferes with chromatin organization after EGA, leading to developmental delay and arrest [37].

We propose that sperm with low active CN and, by extrapolation, a lower amount of rRNA after EGA, may have an impact on functional nuclear organization, ribosome biogenesis and the mass production of proteins during early development. Of course, a lower active CN in sperm may be compensated by a higher CN in the oocyte and there is a huge variation in active CN between individuals/embryos. Therefore, consistent with a multifactorial disease model, the effect of low-sperm rDNA CN on the chances to conceive a pregnancy may be small.

### 3.3. Conclusions and Limitations

High levels of rDNA activity and ribosome biogenesis are required for the efficient production of proteins during early embryogenesis. Although it is plausible to assume that sperm rDNA copies with a hypomethylated promoter are ready for transcription in early embryos, this is difficult to prove by functional experiments. In general, genes with a hypomethylated promoter are transcriptionally active [38]. However, in this context, it is important to mention that rRNA expression is not only regulated by rDNA methylation but also by chromatin alterations, modifications of histones, RNA polymerase I factors, and lncRNAs [39,40].

Our study links sperm rDNA CN with the establishment of an IVF/ICSI pregnancy. Of course, ART outcome does not only depend on male but also even more so on female factors. Unfortunately, we do not have information on the female side, which makes it difficult to conclude that the observed effects are due to the studied sperm parameters.

Overall sperm samples with abnormal semen parameters or without IVF/ICSI pregnancy exhibit a lower number of presumably active hypomethylated rDNA copies. This is explained by promoter hypermethylation in the ASP group and/or a reduced absolute CN in the NSP group. On average, men with fertility problems have 10 hypomethylated, presumably active copies less than men with normozoospermic samples and/or successful IVF/ICSI treatment. Since all samples are from men undergoing IVF/ICSI treatment, low presumably active CN may be associated with poor semen parameters and/or poor pregnancy outcome in this group; however, we do not know whether this also holds true for men without fertility problems. Considering the enormous CN variation between individuals, the power of active CN to predict the fertility status or IVF/ICSI outcome is low. Active CN is only one of many factors contributing to male infertility.

## 4. Materials and Methods

### 4.1. Study Samples

This retrospective study on human sperm samples was approved by the ethics committee at the medical faculty of the University of Würzburg (no. 117/11 and 212/15). The semen samples were collected at the Fertility Center Wiesbaden. After IVF/ICSI treatment, the left-over swim-up sperm fraction (excess material) was pseudonymized, and snap-frozen at −80 °C until further use. To eliminate contamination by bacteria, lymphocytes, epithelial, and other somatic cells, the swim-up sperm samples were gently thawed and further purified by density gradients PureSperm 80 and 40 (Nidacon, Mölndal, Sweden).

A total of 190 sperm samples (Appendix A) was used for rDNA CN counting: 94 were from men with normal and 96 with abnormal parameters using the reference values of the 5th edition of the WHO laboratory manual [41]. Establishment of a pregnancy was determined by biochemical parameters and fetal heart beats: 100 sperm samples led to a pregnancy after IVF/ICSI and 89 did not. For one sample, pregnancy information was not available.

For DNA isolation, the purified sperm cells were resuspended in 300 µL buffer (5 mL of 5 M NaCl, 5 mL of 1 M Tris-HCl; pH 8, 5 mL of 10% SDS; pH 7.2, 1 mL of 0.5 M EDTA; pH 8, 1 mL of 100% β-mercaptoethanol, and 33 mL H_2_O) and 100 µL (20 mg/mL; 600 mAU/mL) proteinase K (Qiagen, Hilden, Germany), and incubated for 2 h at 56 °C. Sperm DNA was isolated using the DNeasy Blood and Tissue kit (Qiagen). DNA concentration and purity were measured by NanoDrop 2000c spectrophotometer (Thermo Fisher Scientific, Waltham, MA, USA). Bisulfite conversion of DNA was performed using the EpiTect Fast 96 Bisulfite kit (Qiagen) and the converted DNA was stored at −20 °C until further use.

### 4.2. Droplet Digital PCR

ddPCR primers (Appendix A) for 28S rDNA and the human TATA-box binding protein (*TBP*) gene (internal reference) were adopted from the literature [42]. ddPCR for absolute rDNA counting was performed according to our previously published protocol [13,16,43]. Briefly, ddPCR was set up as a CNV analysis using the QuantaSoft software version 1.7.4.0917 (Bio-Rad, Feldkirchen, Germany). The FAM fluorescent label was used for the reference gene and the HEX label for rDNA. Droplets were generated using the QX200 droplet generator (Bio-Rad). A QX200 droplet reader (Bio-Rad) was used to detect signals from individual droplets. CN was determined by calculating the ratio of the target molecule concentration to the reference molecule concentration (in copies/µL), multiplied by the CN of the reference gene in the diploid genome. Thresholds separating FAM-positive, HEX-positive, double positive and double negative droplets were manually adjusted using the QX Manager software version 1.2 Standard Edition 2 (Bio-Rad).

### 4.3. Deep Bisulfite Sequencing

DBS primers (Appendix A) were designed for the human upstream control element and core promoter region [13,16,43], containing 25 contiguous CpGs and an A/G variant (GRCh38; chr21:8.205.892) [44]. Overall, >95% of reads represented the major G allele. To avoid the effect of genetic variation on our methylation results, only reads representing the major variant were used for downstream analyses.

rDNA library preparation was performed exactly as described in our previous studies on rDNA CN variation [13,16,43]. Following Illumina’s (San Diego, CA, USA) recommendations for sequencing low-diversity libraries with the NextSeq 200, the rDNA library was spiked with 25% PhiX. Sequencing with the Reagent Kit P1 (300 cycles) (Illumina) yielded 150 bp paired-end reads, which were processed by the Illumina BCL Convert software version 4.2.7. Only reads with an overall bisulfite conversion rate > 95% were considered for downstream processing of FASTQ files using Amplikyzer2 software (https://bitbucket.org/svenrahmann/amplikyzer/wiki/Home, accessed on 22 September 2025) [45].

### 4.4. Statistical Analysis

Both descriptive and inferential statistical analyses were conducted using IBM SPSS version 28. The number of rDNA copies within a given methylation range (i.e., presumably active copies with ≤10% methylation) was calculated by multiplying the absolute CN (from ddPCR) with the percentage of reads (from DBS) in the corresponding methylation bin (i.e., 0–10%). The Kolmogorov–Smirnov and Shapiro–Wilk tests were used to determine data distribution. Group comparisons, based on data distribution, were conducted using non-parametric Mann–Whitney U tests. Spearman’s correlations were used to determine the relationship between two continuous variables, such as active CN and different semen parameters. *p* values ≤ 0.05 were considered statistically significant.

## Figures and Tables

**Figure 1 ijms-26-10657-f001:**
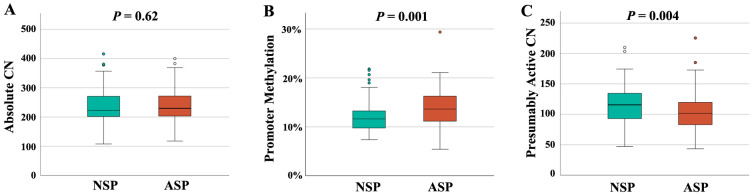
rDNA CN and methylation in NSP (*N* = 94) and ASP (*N* = 96) samples. (**A**) There is no between-group difference in absolute CN. (**B**) rDNA promoter methylation is significantly (*p* = 0.001) lower in the NSP group. (**C**) The number of hypomethylated (≤10%), presumably active copies is significantly (*p* = 0.004) higher in the NSP group. The median is presented by a horizontal line. The bottom of the box indicates the 25th and the top the 75th percentile. Outliers are indicated by open circles.

**Figure 2 ijms-26-10657-f002:**
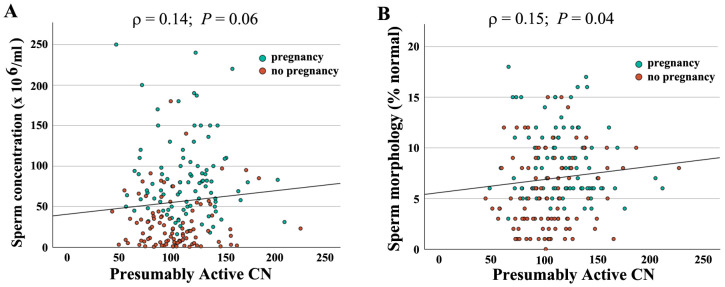
Correlation of presumably active rDNA CN with semen parameters. Presumably active CN ranging from 43 to 226 is indicated on the X axis, sperm concentration (**A**) and morphology (**B**) on the Y axis. Green dots represent samples leading to IVF/ICSI pregnancy and red dots samples without pregnancy. Both sperm concentration (*ρ* = 0.14, *p* = 0.06) and morphology (*ρ* = 0.15, *p* = 0.04) are positively correlated with presumably active CN. Please note that overall samples leading to pregnancy have better semen parameters.

**Figure 3 ijms-26-10657-f003:**
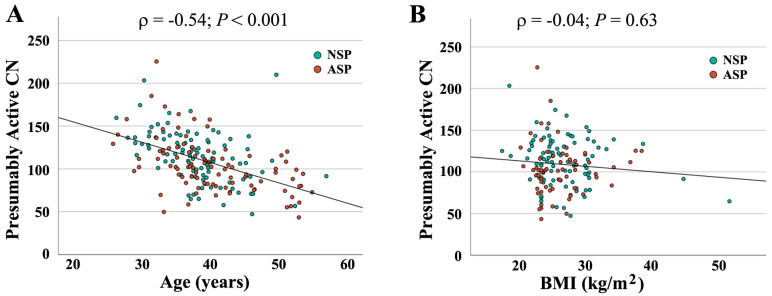
Correlation of presumably active rDNA CN with donor age (**A**) and BMI (**B**). Age and BMI, respectively, are indicated on the X axis, presumably active CN on the Y axis. Green dots (*N* = 94) represent samples with normal and red dots (*N* = 96) with an abnormal spermiogram. Presumably active CN significantly decreases with donor age (*ρ* = −0.54, *p* < 0.001) but is not affected by BMI (*ρ* = −0.04, *p* = 0.63).

**Figure 4 ijms-26-10657-f004:**
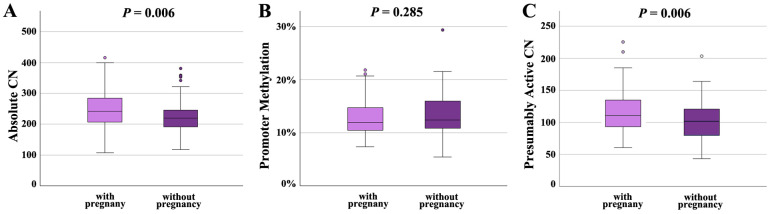
rDNA CN and methylation in sperm samples leading to a pregnancy (*N* = 100) and samples not leading to a pregnancy (*N* = 89). (**A**) Absolute CN is significantly (*p* = 0.006) lower in samples without pregnancy. (**B**) There is no between-group difference in the promoter methylation level. (**C**) The number of hypomethylated (≤10%), presumably active copies is significantly (*p* = 0.006) lower in the group without pregnancy. The median is presented by a horizontal line. The bottom of the box indicates the 25th and the top the 75th percentile. Outliers are indicated by open circles.

**Figure 5 ijms-26-10657-f005:**
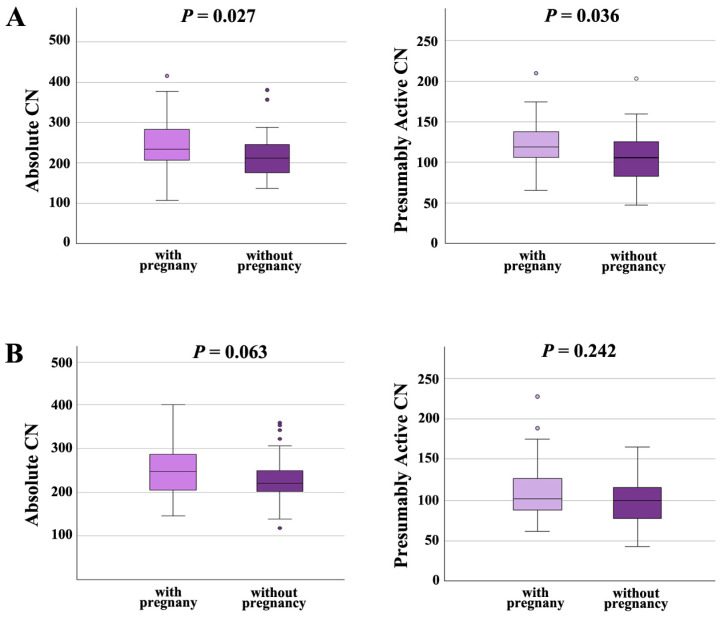
(**A**) Absolute and presumably active rDNA CN in men with normal semen parameters are significantly higher in samples leading to pregnancy than in samples that did not. (**B**) In men with abnormal semen parameters, absolute and active rDNA are also higher in samples with pregnancy; however, there is only a trend or no significant between-group difference. The median is presented by a horizontal line. The bottom of the box indicates the 25th and the top the 75th percentile. Outliers are indicated by open circles.

**Table 1 ijms-26-10657-t001:** rDNA CN, methylation, and clinical parameters of NSP samples (*N* = 94).

	Mean ± SD	Median	Range
**Absolute Copy Number**	236 ± 61	223	108–416
**Presumably Active Copy Number**	115 ± 31	116	47–210
**Methylation (%)**	12.1 ± 3.2	11.7	7.3–23.2
**Age**	38.1 ± 5.6	37.8	26.2–56.9
**BMI (kg/m^2^)**	26.7 ± 5.0	25.7	17.5–51.7
**Volume (mL)**	3.3 ± 1.3	3.0	1.4–8.0
**Concentration (×10^6^/mL)**	85 ± 50	78.0	16–2 50
**Morphology (% normal)**	8.5 ± 3.5	8.0	3–18
**Motility (% motile)**	58 ± 11	59	34–85

**Table 2 ijms-26-10657-t002:** rDNA CN, methylation, and clinical parameters of ASP samples (*N* = 96).

	Mean ± SD	Median	Range
**Absolute Copy Number**	240 ± 56	230	118–400
**Presumably Active Copy Number**	104 ± 31	101	43–226
**Methylation (%)**	13.9 ± 3.6	13.7	5.4–29
**Age**	40 ± 8.6	39.1	25.8–54.8
**BMI (kg/m^2^)**	26.2 ± 3.9	25.1	20.3–38.5
**Volume (mL)**	3.3 ± 1.5	3.3	0.2–9.0
**Concentration (×10^6^/mL)**	29 ± 32	18.0	0.5–180
**Morphology (% normal)**	5.5 ± 3.7	5.0	0–15
**Motility (% motile)**	36 ± 17.5	33.0	1–88

**Table 3 ijms-26-10657-t003:** rDNA CN, methylation, and clinical parameters of samples with pregnancy (*N* = 100).

	Mean ± SD	Median	Range
**Absolute Copy Number**	249 ± 62	243	108–416
**Presumably Active Copy Number**	115 ± 31	110	61–226
**Methylation (%)**	12.7 ± 3.2	12.0	7.3–24.0
**Age**	39.1 ± 6.5	38.2	25.8–57.0
**BMI (kg/m^2^)**	26.6 ± 4.1	25.2	20.3–44.8
**Volume (mL)**	3.3 ± 1.5	3.0	0.8–9.0
**Concentration (×10^6^/mL)**	64 ± 47	56.0	1.0–200
**Morphology (% normal)**	8.1 ± 4.0	8.0	1.0–18
**Motility (% motile)**	49.0 ± 15.6	50.0	1.0–85

**Table 4 ijms-26-10657-t004:** rDNA CN, methylation, and clinical parameters of samples without pregnancy (*N* = 89).

	Mean ± SD	Median	Range
**Absolute Copy Number**	225 ± 52	220	118–381
**Presumably Active Copy Number**	103 ± 30	102	43–203
**Methylation (%)**	13.3± 3.8	12.5	5.4–29.0
**Age**	39.3± 6.4	38.0	26.2–54.8
**BMI (kg/m^2^)**	26.4 ± 4.9	25.5	17.5–51.7
**Volume (mL)**	3.3 ± 1.2	3.2	0.2–6.5
**Concentration (×10^6^/mL)**	48 ± 52	36.0	0.5–250
**Morphology (% normal)**	5.8 ± 3.4	6.0	0–15
**Motility (% motile)**	43.5 ± 21.0	45.0	5–88

## Data Availability

The original contributions presented in this study are included in the article and Appendix A. Further inquiries can be directed to the corresponding author.

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
