# Peer review of "Sperm rDNA Copy Number and Methylation Are Associated with Male-Factor Infertility"

_ijms, 2025, doi:10.3390/ijms262110657_

Round 1

Reviewer 1 Report

Comments and Suggestions for Authors

In their manuscript titled "Sperm rDNA Copy Number and Methylation are Associated with Male-Factor Infertility," Michler et al. have analyzed the total and active (hypomethylated) rDNA CNs in DNA obtained from human sperm samples collected at a fertility center. After performing appropriate statistical analyses, they have determined that decreased active rDNA CNs are associated with abnormal sperm parameters and lower rates of clinical pregnancy after IVF/ICSI. While their results are a valuable contribution to the field, I have several comments which should be addressed prior to publication.

Major comments:

1) The introduction is nicely written and gives a brief overview of the most important points. I suggest separating the last part of the introduction (lines 61-63) as a separate paragraph at the end and structuring it as follows - 1 sentence about your hypothesis (males with fertility problems having lower absolute/active rDNA CNs), 1 sentence about your aim (what you're comparing and how you're measuring it), and 1 sentence about the possible implications of your findings on future studies or clinical outcomes (e.g. Our findings could contribute to the existing knowledge about the role of ... and can be a basis for future studies on ...)

2) You should add a supplementary table containing the absolute and active rDNA CN for each sample, along with their clinical and spermiogram paramteres (age, BMI, spermatozoa concentration, etc.). For ASP samples, you should ephasize how the sperm was abnormal (oligozoospermia, teratoasthenozoospermia, etc.). 

3) Lines 86-90: Was there any correlation between active rDNA CN and sperm motility? If not, this should be stated. Also, have you performed these same correlation analyses for absolute rDNA CN? If you have, and none are statistically significant, this should also be stated in the results.

4) I suggest performing an addition analysis on ASP subgroups (oligozoospermia, teratozoospermia, asthenozoospermia) vs NSP to confirm the trends between promoter methylation / active rDNA CN and specific (abnormal) sperm paramters. Basically the same as Figure 1, except instead of NSP vs ASP it would be NSP vs oligozoospermia, NSP vs teratozoospermia, and NSP vs asthenozoospermia.

5) You must add more details about statistical analysis - which test was used to assess distribution of data, which test was used to compare groups, which test was used for correlation analysis, what p-value was considered significant, etc. After adding this to the methods, you can remove test names from the results section.

Minor comments:

1) Line 38: you should state the full name for AZF, azoospermia factor.

2) Line 51: Acrocentric short arms should be changed to "short arms of acrocentric chromosomes."

3) Line 50: Add (TU) after transcription units.

4) Line 57: I suggest changing "soma" to "somatic cells"

5) Line 60: I suggest combining the two statements "... paternal age, and this is associated with increasing rDNA methylation [14,16,19,20]."

6) Line 263: Even though the protocol is given in detail in the referenced studies, you should at least name the devices (manufacturer) used for droplet generation and signal detection, and the software + version used for analysis.

7) Line 268: "in human sperm" is extra and can be removed.

8) Lines 127-131: You should remove the % as these are CNs (absolute values). The last statement is too strong for this level of a study; you cannot determine (or even approximate) the minimum number of active copies required to establish pregnancy, because maybe someone can establish a pregnancy with 50 active copies. You can, however, determine that "approximately 60 active copies are sufficient to establish a pregnancy," which is my suggestion for the statement. Line 165, required should also be changed to sufficient.

9) Line 207: I suggest adding "might" in front of "have and impact".

10) Lines 217-220 should be moved to the introduction as they state just general information.

Author Response

In their manuscript titled "Sperm rDNA Copy Number and Methylation are Associated with Male-Factor Infertility," Michler et al. have analyzed the total and active (hypomethylated) rDNA CNs in DNA obtained from human sperm samples collected at a fertility center. After performing appropriate statistical analyses, they have determined that decreased active rDNA CNs are associated with abnormal sperm parameters and lower rates of clinical pregnancy after IVF/ICSI. While their results are a valuable contribution to the field, I have several comments which should be addressed prior to publication.

Major comments:

1) The introduction is nicely written and gives a brief overview of the most important points. I suggest separating the last part of the introduction (lines 61-63) as a separate paragraph at the end and structuring it as follows - 1 sentence about your hypothesis (males with fertility problems having lower absolute/active rDNA CNs), 1 sentence about your aim (what you're comparing and how you're measuring it), and 1 sentence about the possible implications of your findings on future studies or clinical outcomes (e.g. Our findings could contribute to the existing knowledge about the role of ... and can be a basis for future studies on ...)

RESPONSE 1:   We have added a paragraph at the end of the introduction, explaining the aim, the experimental approach and the possible implications of our study.

2) You should add a supplementary table containing the absolute and active rDNA CN for each sample, along with their clinical and spermiogram paramteres (age, BMI, spermatozoa concentration, etc.). For ASP samples, you should emphasize how the sperm was abnormal (oligozoospermia, teratoasthenozoospermia, etc.).

RESPONSE 2:  We have added a supplementary table S1 with the requested information.

3) Lines 86-90: Was there any correlation between active rDNA CN and sperm motility? If not, this should be stated. Also, have you performed these same correlation analyses for absolute rDNA CN? If you have, and none are statistically significant, this should also be stated in the results.

RESPONSE 3:   In the results we state that when considering all sperm samples, absolute CN was not significantly correlated with semen parameters (concentration: r = -0.04, P = 0.56; motility: r = -0.10, P = 0.18); morphology: r = -0.06, P = 0.45). In contrast, there was a significant correlation of active CN with sperm concentration (r = 0.14, P = 0.05) and morphology (r = 0.16, P = 0.03), but not with motility (r = 0.10, P = 0.15).

4) I suggest performing an addition analysis on ASP subgroups (oligozoospermia, teratozoospermia, asthenozoospermia) vs NSP to confirm the trends between promoter methylation / active rDNA CN and specific (abnormal) sperm parameters. Basically the same as Figure 1, except instead of NSP vs ASP it would be NSP vs oligozoospermia, NSP vs teratozoospermia, and NSP vs asthenozoospermia.

RESPONSE 4:   To corroborate the association between promoter methylation/active CN and abnormal semen parameters, we have compared NSP samples with different ASP subgroups (Table S2). Most ASP samples exhibited oligoasthenoteratozoospermia (OAT) (N = 37) or asthenozoospermia (reduced motility) (N = 49). There was a significant or trend difference in methylation (OAT: P = 0.023; asthenozoospermia: P = 0.002) and presumably active CN (P = 0.058 and 0.032, respectively), compared to the NSP group. Despite small sample sizes, significant/trend between-group differences were also observed for oligozoospermia (low sperm count), cryptozoospermia (more severe variant of oligozoospermia), and teratozoospermia (high percentage of abnormally shaped sperm). As expected, absolute CN did not differ between NSP and ASP subgroups (Table S2).

5) You must add more details about statistical analysis - which test was used to assess distribution of data, which test was used to compare groups, which test was used for correlation analysis, what p-value was considered significant, etc. After adding this to the methods, you can remove test names from the results section.

RESPONSE 5:   In the last paragraph of Materials and Methods we provide more information on which tests have been used for statistical analyses.

Minor comments:

1) Line 38: you should state the full name for AZF, azoospermia factor.

2) Line 51: Acrocentric short arms should be changed to "short arms of acrocentric chromosomes."

3) Line 50: Add (TU) after transcription units.

4) Line 57: I suggest changing "soma" to "somatic cells"

5) Line 60: I suggest combining the two statements "... paternal age, and this is associated with increasing rDNA methylation [14,16,19,20]."

6) Line 263: Even though the protocol is given in detail in the referenced studies, you should at least name the devices (manufacturer) used for droplet generation and signal detection, and the software + version used for analysis.

7) Line 268: "in human sperm" is extra and can be removed.

8) Lines 127-131: You should remove the % as these are CNs (absolute values). The last statement is too strong for this level of a study; you cannot determine (or even approximate) the minimum number of active copies required to establish pregnancy, because maybe someone can establish a pregnancy with 50 active copies. You can, however, determine that "approximately 60 active copies are sufficient to establish a pregnancy," which is my suggestion for the statement. Line 165, required should also be changed to sufficient.

9) Line 207: I suggest adding "might" in front of "have and impact".

10) Lines 217-220 should be moved to the introduction as they state just general information.

RESPONSE TO MINOR COMMENTS:  All suggested changes have been incorporated in the revised manuscript. Thank you carefully reading and improving our manuscript.

Reviewer 2 Report

Comments and Suggestions for Authors

It is an original paper, with descriptive data, but quite novel and offering new perspectives in the field of human male infertility.

This paper addressed originally one specific aspect of male infertility, which has been a major subject of the team members which accumulates paper on the subject. The last was published In IJMS in 2025 (Potabattula, R.; Dittrich, M.; Hahn, T.; Schorsch, M.; Ptak, G.E.; Haaf, T. rDNA copy number variation and methylation in human and mouse sperm. Int. J. Mol. Sc.i 2025, 26, 4197). In the present study, a connection between rDNA copy number/methylation is tentatively established with infertility in humans, which is here the original angle.

The study is based upon 94 controls and 96 abnormal sperm carrier. Mild but significant differences were found for rDNA promoter methylation (higher in abnormal sperm), and the number of hypomethylated copies lower in this category (two consistent observations). Marginally significant correlations were found between active copy number and sperm concentration/morphology, whiule interestingly the number of active copy number decreases with age (but not BMI). This information is interesting and novel.

Another classification is performed in normal and abnormal sperm parameter showing trends

An association is shown between the results (pregnancy and non-pregnancy) and the decrease number in the later case and active copy number.

The only minor remark that I have is the following:

The hypomethylation is correlated in the paper with the presumably active status of the promoter. This association is not a causality as mentioned peremptorily in the abstract ‘The loss of active rDNA copies is explained by an increased promoter methylation (13.9% in ASP vs. 12.1% in NSP)’. I suggest the authors be more careful throughout the text. There are cases where hypermethylation is not associated with silencing, even when in gene promoters.

Author Response

It is an original paper, with descriptive data, but quite novel and offering new perspectives in the field of human male infertility.

This paper addressed originally one specific aspect of male infertility, which has been a major subject of the team members which accumulates paper on the subject. The last was published In IJMS in 2025 (Potabattula, R.; Dittrich, M.; Hahn, T.; Schorsch, M.; Ptak, G.E.; Haaf, T. rDNA copy number variation and methylation in human and mouse sperm. Int. J. Mol. Sc.i 2025, 26, 4197). In the present study, a connection between rDNA copy number/methylation is tentatively established with infertility in humans, which is here the original angle.

The study is based upon 94 controls and 96 abnormal sperm carrier. Mild but significant differences were found for rDNA promoter methylation (higher in abnormal sperm), and the number of hypomethylated copies lower in this category (two consistent observations). Marginally significant correlations were found between active copy number and sperm concentration/morphology, while interestingly the number of active copy number decreases with age (but not BMI). This information is interesting and novel.

Another classification is performed in normal and abnormal sperm parameter showing trends

An association is shown between the results (pregnancy and non-pregnancy) and the decrease number in the later case and active copy number.

The only minor remark that I have is the following:

The hypomethylation is correlated in the paper with the presumably active status of the promoter. This association is not a causality as mentioned peremptorily in the abstract ‘The loss of active rDNA copies is explained by an increased promoter methylation (13.9% in ASP vs. 12.1% in NSP)’. I suggest the authors be more careful throughout the text. There are cases where hypermethylation is not associated with silencing, even when in gene promoters.

RESPONSE:   In  the Limitations we state clearly that rRNA expression is not only regulated by rDNA methylation, but also by chromatin alterations, modifications of histones, RNA polymerase I factors, and lncRNAs. Throughout the manuscript including figures and tables, active CN was replaced by presumably active CN.